# Divergent engagements between adeno-associated viruses with their cellular receptor AAVR

Ran Zhang[1,2,10], Guangxue Xu[1,10], Lin Cao[3,10], Zixian Sun[2,10], Yong He[3], Mengtian Cui[4], Yuna Sun[5], Shentao Li[4], Huapeng Li[6], Lan Qin[7], Mingxu Hu[2], Zhengjia Yuan[8], Zipei Rao[9], Wei Ding[4], Zihe Rao[1,2,3,5] & Zhiyong Lou [1]

Adeno-associated virus (AAV) receptor (AAVR) is an essential receptor for the entry of multiple AAV serotypes with divergent rules; however, the mechanism remains unclear. Here, we determine the structures of the AAV1-AAVR and AAV5-AAVR complexes, revealing the molecular details by which PKD1 recognizes AAV5 and PKD2 is solely engaged with AAV1. PKD2 lies on the plateau region of the AAV1 capsid. However, the AAV5-AAVR interface is strikingly different, in which PKD1 is bound at the opposite side of the spike of the AAV5 capsid than the PKD2-interacting region of AAV1. Residues in strands F/G and the CD loop of PKD1 interact directly with AAV5, whereas residues in strands B/C/E and the BC loop of PKD2 make contact with AAV1. These findings further the understanding of the distinct mechanisms by which AAVR recognizes various AAV serotypes and provide an example of a single receptor engaging multiple viral serotypes with divergent rules.

[1] MOE Key Laboratory of Protein Science & Collaborative Innovation Center of Biotherapy, School of Medicine, Tsinghua University, Beijing 100084, China. [2] School of Life Sciences, Tsinghua University, Beijing 100084, China. [3] State Key Laboratory of Medicinal Chemical Biology, College of Life Science and College of Pharmacy, Nankai University, Tianjin 300071, China. [4] Department of Biochemistry and Molecular Biology, School of Basic Medical Sciences, Capital Medical University, Beijing 100069, China. [5] National Laboratory of Macromolecules, Institute of Biophysics, Chinese Academy of Science, Beijing 100101, China. [6] PackGene Biotech, Guangzhou 510000 Guangdong, China. [7] DIAN Diagnostics, Hangzhou 300244 Zhejiang, China. [8] High School Attached to Capital Normal University, Beijing 100048, China. [9] No. 4 High School, Beijing 100034, China. [10] These authors contribute equally: Ran Zhang, Guangxue Xu, Lin Cao, Zixian Sun. Correspondence and requests for materials should be addressed to Z.L. (email: louzy@mail.tsinghua.edu.cn)

Adeno-associated viruses (AAVs) belong to the *Dependoparvovirus* genus within the Parvoviridae family[1]. Because of the broad differences in their tissue tropism and transduction efficiency and the absence of pathology, AAVs are the promising vehicles for therapeutic gene delivery[1,2]. To date, over 100 natural AAV variants have been identified from various hosts and tissues, and their tropism characteristics have emerged as one of the important features for their potential in clinical development[3]. The *Dependoparvovirus* genus diverges into two monophyletic groups, in which one contains clades that are specific to humans (clades A, B, and C) and another comprises a mixture of clades that were isolated exclusively from humans (clade F), exclusively from nonhuman primates (clade D), or from both human and nonhuman primates (clade E)[4,5]. The representative serotypes include AAV1/6 (clade A), AAV2 (clade B), an AAV2–3 hybrid (clade C), AAV7 (clade D), AAV8 (clade E), and AAV9 (clade F), while AAV3, AAV4, and AAV5 are assigned as individual clones[4,5].

Naturally occurring AAVs utilize glycan moieties for their initial attachment to the cell surface, e.g., heparan sulfate proteoglycans (HSPGs) for AAV2/3/6, N-terminal galactose for AAV9, and sialic acid (SIA) moieties for AAV1/4/5/6 (refs. [6–8]). Adeno-associated virus receptor (AAVR), which is a glycosylated protein containing five polycystic kidney disease (PKD) repeat domains in its extracellular portion[9], was recently identified as a key proteinaceous receptor for multiple AAV serotypes to employ the post-attachment events of viral entry[10]. AAV serotypes have evolved to engage in distinct interactions with the same receptor, AAVR[10,11]; AAV2 (clade B) predominantly interacts with PKD2, and AAV5 (an individual clone) makes contact with PKD1 (refs. [10,11]). Other serotypes (including AAV1 and AAV8) require a combination of PKD1 and PKD2 for efficient viral transduction, but both AAV1 and AAV8 bind to only PKD2 in the virus overlay assay[11]. A notable observation is that AAVR is a glycoprotein, but its N-linked glycosylation is not strictly required for AAV2 transduction[11].

In a previous work, we determined the structure of the AAV2-AAVR complex at atomic resolution and elucidated the molecular mechanism by which AAV2 attaches to AAVR through PKD2 (ref. [12]). To dissect the divergent rules of engagements between multiple AAV serotypes with AAVR, we select two AAV serotypes, AAV1 and AAV5, and solve the atomic structures of their complexes with AAVR by cryo-EM. The cryo-EM maps show that AAVR PKD2 alone (without PKD1) contacts AAV1 being similar to the AAV2-AAVR complex, while PKD1 directly binds at a completely distinct position on the AAV5 capsid. These results further the understanding of AAV-AAVR recognition at the atomic level and provide an example of one receptor engaging multiple viral strains with divergent rules.

## Results

### Architecture of the AAV1-AAVR and AAV5-AAVR complexes.
The AAV1 and AAV5 particles were incubated with extracellular segments of AAVR containing PKD1–5 to form the virus–receptor complexes. Meanwhile, the AAV1 and AAV5 particles alone were treated under identical conditions to obtain the unbound structures to identify potential conformational shifts upon receptor binding. The cryo-EM micrographs indicate the presence of AAVR on the AAV1 and AAV5 capsids, but at distinct positions (Fig. 1). The final resolutions of the cryo-EM reconstruction were estimated by the FSC 0.143 cutoff to be 3.07 Å for AAV1 alone, 3.30 Å for the AAV1-AAVR complex, 3.18 Å for AAV5 alone, and 3.18 Å for the AAV5-AAVR complex (Supplementary Fig. 1, Supplementary Table 1).

We previously showed that PKD2 binds to the AAV2 capsid[12] (Fig. 1c, d). Similarly, although the entire ectodomain of AAVR containing five PKDs was incubated with AAV1 and AAV5, the extra densities attached to the virus capsids each correspond to one PKD domain (Fig. 1a, b, e, f, Supplementary Fig. 2). The density maps at atomic resolution, in particular the density at the virus–receptor interface, revealed that PKD2 binds to AAV1 and PKD1 binds to AAV5 (Supplementary Figs. 2 and 3). In the AAV5-AAVR complex, the side chains of $_{AAVR}$H351 and $_{AAVR}$Y355 in PKD1 have featured large densities comparing to their counterparts $_{AAVR}$G449/$_{AAVR}$E454 in PKD2 and $_{AAVR}$S543/$_{AAVR}$K547 in PKD3. (The residues are numbered as they are in full-length AAVR with the accession number NP_079150.3.) (Supplementary Fig. 3a, b). Moreover, the side chain of $_{AAVR}$S356 in PKD1 is also remarkably smaller than those of $_{AAVR}$K455 in PKD2 and $_{AAVR}$V548 in PKD3. These features gave us confidence that the density bound to AAV5 corresponds to PKD1. Meanwhile, the extra density attached to AAV1 is identical to that attached to the AAV2 capsid, suggesting that PKD2 contacts AAV1 (Supplementary Fig. 3a, c). Previous works have shown that though PKD1 plays a critical role and PKD2 plays an additional role in AAV1 transduction, AAV1 binds only to PKD1 in the virus overlay assay[11]. Consistently, we did not observe additional density corresponding to PKD1 in the AAV1-AAVR complex, suggesting that PKD1 may affect AAV1 infection via another mechanism rather than through direct interaction with the virus.

Similar to the binding of PKD2 to the AAV2 capsid, PKD2 lies on the plateau region of the AAV1 capsid and interacts with the inner rim of the spike surrounding the icosahedral threefold axis (Fig. 1a–d). In contrast, PKD1 contacts the outer rim of the spike on the AAV5 capsid outwards the icosahedral threefold axis, which is on the opposite side of AAVR PKD2 bound to AAV1/AAV2, and spans the canyon region towards the icosahedral fivefold axis and the twofold/fivefold wall of the capsid (Fig. 1e, f).

### Structure of the AAV5-AAVR complex.
The AAVR PKD1 polypeptide spanning residues from $_{AAVR}$V305 to $_{AAVR}$E401 was built into the density and refined (Supplementary Fig. 4). PKD1 adopts an Ig-like fold and consists of two stacked antiparallel β-sheets containing nine strands. The left half contains strands A-A′-G-F-C, and the right half is composed of strands B-B′-E-D, which are held together by a hydrophobic core of buried side chains.

Each PKD1 molecule mainly contacts one AAV5 capsomer (named capsomer A) but makes one additional contact with an adjacent AAV5 capsomer (named capsomer B) (Fig. 2a, b, Supplementary Figs 2a, Supplementary Table 3). PKD1 spans over the canyon region of the AAV5 capsid like a bridge. The PKD1 residues in strands F/G and the CD loop interact with the AAV5 capsomers to form a major fulcrum (Fig. 2c–e, Supplementary Fig. 4). Additionally, the N-terminal residue, $_{AAVR}$V305, forms a non-covalent interaction with $_{AAV5}$S319 of capsomer A, acting as a minor fulcrum to support the spanning of PKD1 over the canyon (Fig. 2c–e, Supplementary Fig. 4).

The interacting residues on AAV5 capsomer A are mainly located in the βGH13-βGH14 loop (including residues $_{AAV5}$S531/Q532/N535/A540-M547), the βI-βI1 loop (residues $_{AAV5}$Q697 and $_{AAV5}$F698), and the region spanning βI2 (residues $_{AAV5}$E708, $_{AAV5}$R710 and $_{AAV5}$T712) (the defined secondary structure of the AAV5 capsid is shown in Supplementary Table 2) (Fig. 2d, e). Together with $_{AAV5}$N443′ from the adjacent AAV5 capsomer B, these regions act as the major fulcrum that supports the spanning of PKD1 over the canyon. Additionally, residue $_{AAV5}$S319 in the βD-βDE1 loop of capsomer A contacts the N-terminal residue of PKD1, $_{AAVR}$V305, and forms the minor fulcrum to stabilize virus–receptor interactions.

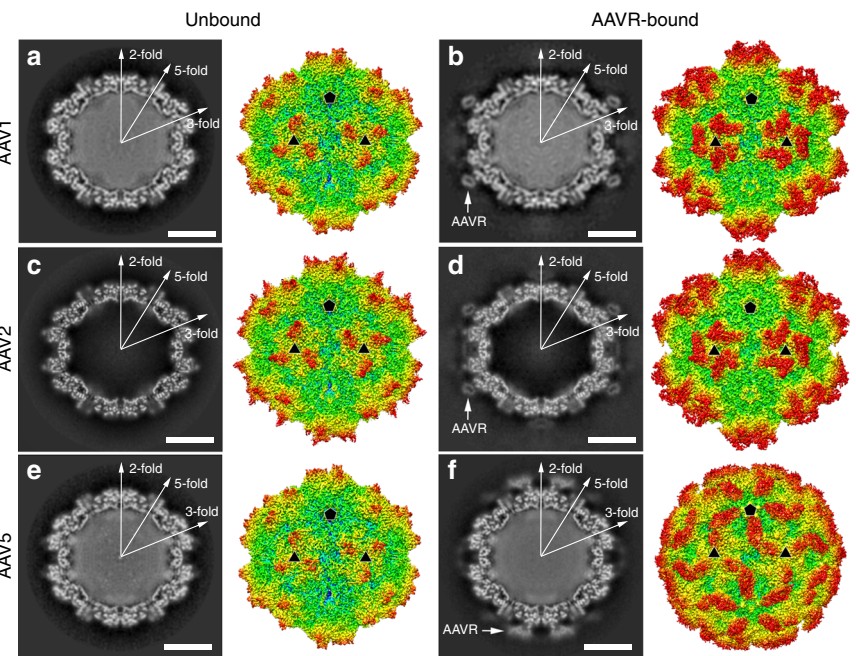

**Fig. 1** The cryo-EM structures. The central cross-sections through the cryo-EM maps and the rendered images of unbound AAV1 (**a**), AAV2 (**c**), and AAV5 (**e**) and AAVR-bound AAV1 (**b**), AAV2 (**d**), AAV5 (**f**) are shown in the left and right halves of each panel, respectively. The central cross-sections are shown with the icosahedral two-, three- and fivefold axes. The scale bars represent 100 Å. Depth cueing is used to indicate the radius by color (<90 Å: blue; 100–125 Å: from cyan to yellow; >140 Å: red). Icosahedral five- and threefold axes are represented by pentagons and triangles. Density for bound AAVRs is indicated by arrows in the central cross-sections

In the structure of the AAV2-AAVR complex, the binding of AAVR PKD2 to the AAV2 capsid results in the remarkable movement of two structural elements at the virus–receptor interface, the βBC1-βC loop and the βGH2-βGH3 loop, to accommodate the bound PKD2 (ref. [12]). However, the structures of AAV5 capsomers with and without PKD1 attached display strict structural homology with an r.m.s.d of 0.27 Å for all Cα atoms from 519 residues, indicating that no obvious structural shift occurs upon AAVR binding (Supplementary Fig. 5a).

**Impact of interacting residues on AAV5 transduction**. To validate the AAV5-AAVR interactions, we mutated the interacting residues in PKD1 and used surface plasmon resonance (SPR) to evaluate their binding affinities of the mutants to AAV5 (Fig. 2f, Supplementary Fig. 6a–j). A total of nine AAVR residues whose side chains make contact with the AAV5 capsid were mutated to alanine residues, including $_{AAVR}$I349, $_{AAVR}$T350, $_{AAVR}$R353, $_{AAVR}$D354, $_{AAVR}$S356, $_{AAVR}$L376, $_{AAVR}$E378, $_{AAVR}$T397, and $_{AAVR}$K399. As a control, the wild-type (wt) AAVR bound to AAV5 with a KD value of 0.287 μM. The $_{AAVR}$I350A, $_{AAVR}$R353A, $_{AAVR}$L376A, and $_{AAVR}$E378A mutations completely abolished the binding of AAVR to AAV5 in vitro. The $_{AAVR}$D354A, $_{AAVR}$S356A, and $_{AAVR}$T397A mutations decreased the binding affinities by 100-fold compared to that observed for wt AAVR, with KD values ranging from 17 to 30 μM, and the $_{AAVR}$K399A mutation reduced the binding affinity to 71.9 μM. In contrast, the $_{AAVR}$I349A mutation increased the KD value to 0.007 μM.

We next obtained AAVR knockdown cells by stably expressing a previously verified shRNA targeting wt AAVR[12]. By the ectopic expression of nine AAVR mutants ($_{AAVR}$I349A, $_{AAVR}$T350A, $_{AAVR}$R353A, $_{AAVR}$D354A, $_{AAVR}$S356A, $_{AAVR}$L376A, $_{AAVR}$E378A, $_{AAVR}$T397A, and $_{AAVR}$K399A) in AAVR-silenced cells, we found that the introduction of the most AAVR mutants significantly attenuated the percentage of AAV5 infection, and the $_{AAVR}$T350A, $_{AAVR}$R353A, and $_{AAVR}$D354A mutations retained

60–70% as wt AAV5 infection (Fig. 2g, Supplementary Fig. 7b). Their effects were consistent with their decreased binding affinities with AAV5 seen in SPR assays. As expected, the $_{AAVR}$I349A mutation increased AAV5 transduction compared to that observed for wt AAVR. We hypothesized that the $_{AAVR}$I349A mutation might affect the overall folding of AAVR and thus promote AAV5 infection.

We also substituted the AAV5 capsid residues on the interface with alanine residues and tested the transduction activities of the mutated viruses. The substitutions at $_{AAV5}$N546, $_{AAV5}$F698, and $_{AAV5}$R710 completely eliminated AAV5 transduction, and the $_{AAV5}$Y542A mutant retained only 15% transduction activity compared to that of the wt AAV5 (Fig. 2h, Supplementary Fig. 7a). These results are consistent with structural observations that these four AAV5 capsid residues form extensive intermolecular contacts with $_{AAVR}$H351, $_{AAVR}$P352, $_{AAVR}$R353, and $_{AAVR}$L376 in PKD1 to stabilize the AAV5-AAVR interaction (Fig. 2c, d, Supplementary Table 3). Moreover, mutated AAV5 containing $_{AAV5}$Q532A and $_{AAV5}$E708A mutations exhibited decreased AAV5 infectivity that was 50% that of wt AAV5. This is also reasonable since $_{AAV5}$Q532A and $_{AAV5}$E708A interact with PKD1 in the same region as the other four AAV5 residues that have the most significant impact on binding. Other mutations including $_{AAV5}$S319A, $_{AAV5}$N443A, $_{AAV5}$N535A, $_{AAV5}$A546G, $_{AAV5}$L543A, $_{AAV5}$Q697A, and $_{AAV5}$T712A produced mild or negligible effects on AAV5 transduction. In the virus overlay assays, the mutated AAV5 containing $_{AAV5}$Q532A, $_{AAV5}$Y542A, $_{AAV5}$N546, $_{AAV5}$F698, $_{AAV5}$E708A, and $_{AAV5}$R710 mutations displayed obviously lowered binding with wt AAVR (Supplementary Fig. 8a), which is similar with their impacts on virus transduction. An exception was $_{AAV5}$S531A, which increased the functional titer to 12208 ± 344 TU μl$^{-1}$ compared to the functional titer of 7752 ± 414 TU μl$^{-1}$ observed with wt AAV5. Because $_{AAV5}$S531A only forms two non-covalent interactions with the Cα and Cβ atoms of $_{AAVR}$R353

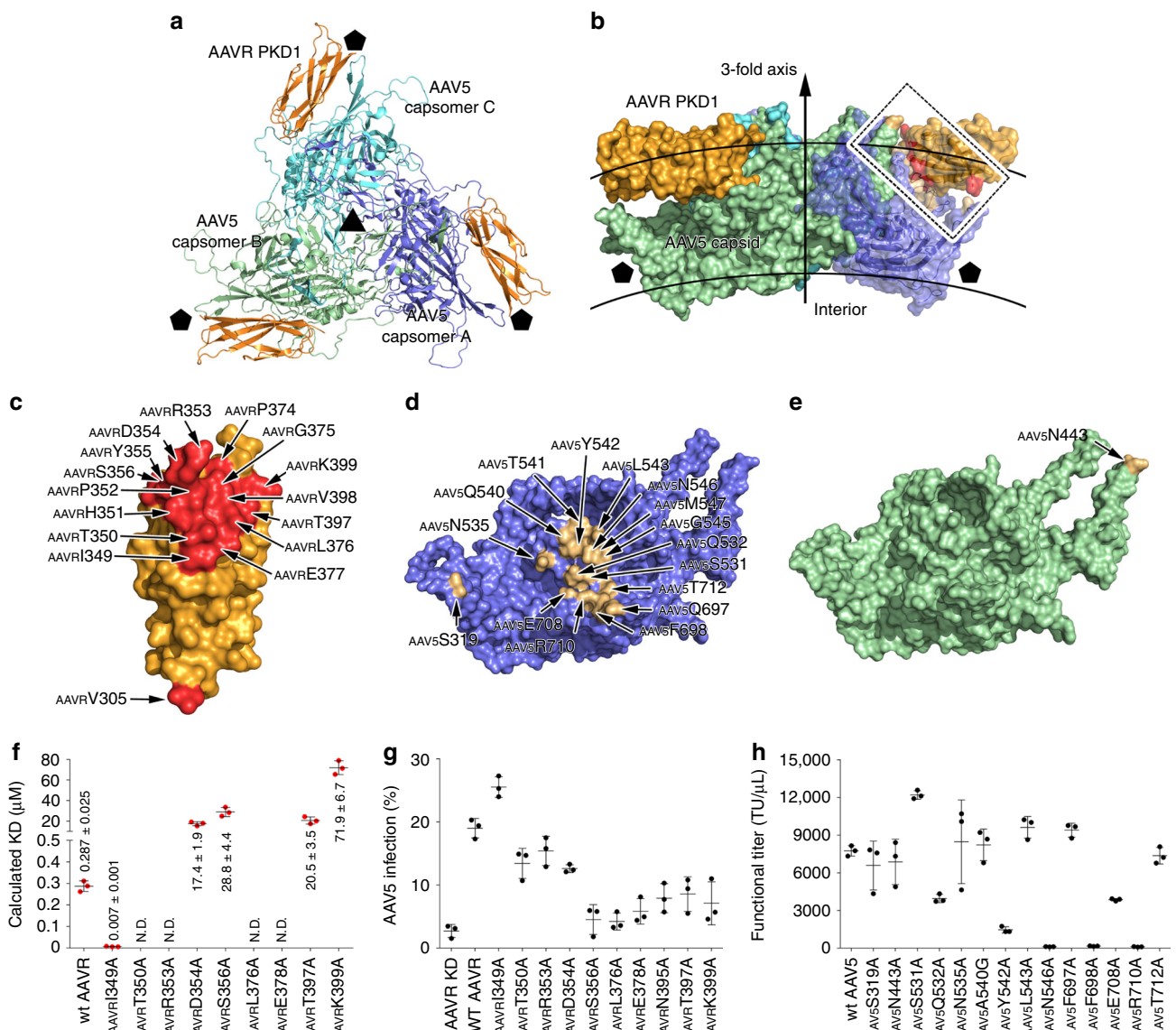

**Fig. 2** AAVR PKD1 binds with AAV5. **a**, **b** The structures of trimeric AAV5 capsomers in complex with AAVR are shown in ribbon representation from a top view (**a**) and as a covered surface from a perpendicular side view (**b**). The rough inner and outer boundaries of the viral capsid are marked with gray arcs in **b**. The three AAV5 capsomers are colored blue, green, and cyan, respectively. Bound AAVRs are shown in orange. The five- and threefold axes are indicated by pentagons and triangles, respectively. The interacting residues on one AAVR PKD1 and the corresponding AAV5 capsomers are colored red and gold, respectively, in **b**, and the interacting region is framed with dashed lines. The residues in PKD1 at the interface are shown in red (**c**); the interface residues in AAV5 capsomer A (**d**), and capsomer B (**e**) are colored gold and labeled. See also Supplementary Table 3. **f** A total of nine AAVR mutants were tested for their ability to bind AAV5 by BIAcore sensorgrams in triplicate experiments. The calculated KD values for the binding of each mutant to AAV5 are summarized as the mean values of three experiments with standard errors. The original curves are presented in Supplementary Fig. 6. **g** The impact of AAVR PKD1 mutants overexpressed in AAVR-silenced HEK293T cells on AAV5 transduction. Cells were transfected with wt AAVR or different AAVR mutants as indicated followed by infection with AAV5-mCherry at an MOI of $3 \times 10^{6}$ vg cell$^{-1}$. The percentage of mCherry-positive cells are plotted as means ± standard errors ($n = 3$). Expression of wt AAVR and mutant AAVRs was evaluated by immunoblot analysis with β-actin as a control (see Supplementary Fig. 13). **h** AAV5 molecules bearing capsid mutations at the receptor binding sites impacted viral transduction. HEK293T cells were infected with serial dilutions of AAV5 mutants starting at an MOI of $3 \times 10^{6}$ vg cell$^{-1}$. Functional titer refers to the number of virus particles that can infect the cell in every microliter. GFP expression was determined 48 h post-transduction. The functional titers are plotted as means ± standard errors ($n = 3$). Source data are provided as a Source Data file

in the AAV5-AAVR complex, these interactions may not be strictly required for the AAV5-AAVR interaction, and the mutation of $_{AAV5}$S531 to an alanine residue may alter the conformation of this region, improving AAV5 infectivity.

**Structure of the AAV1-AAVR complex.** Like that found in the AAV2-AAVR complex, the extra density on the AAVR-bound

AAV1 particle was identified as PKD2 (Supplementary Fig. 3a, c). The interaction of PKD2 with the AAV1 capsid is homologous to that in the AAV2-AAVR complex. Each PKD2 molecule contacts two AAV1 capsomers (Fig. 3a, b, Supplementary Table 5). Residues in strands B/C/E and the BC loop of PKD2 interact with the AAV1 capsid (Fig. 3c). The AAVR-interacting residues in one AAV2 capsomer (named capsomer A) are mainly located in the βBC2-βC loop, βEF2-αEF3 loop, βGH6-αGH6′ loop and αGH5

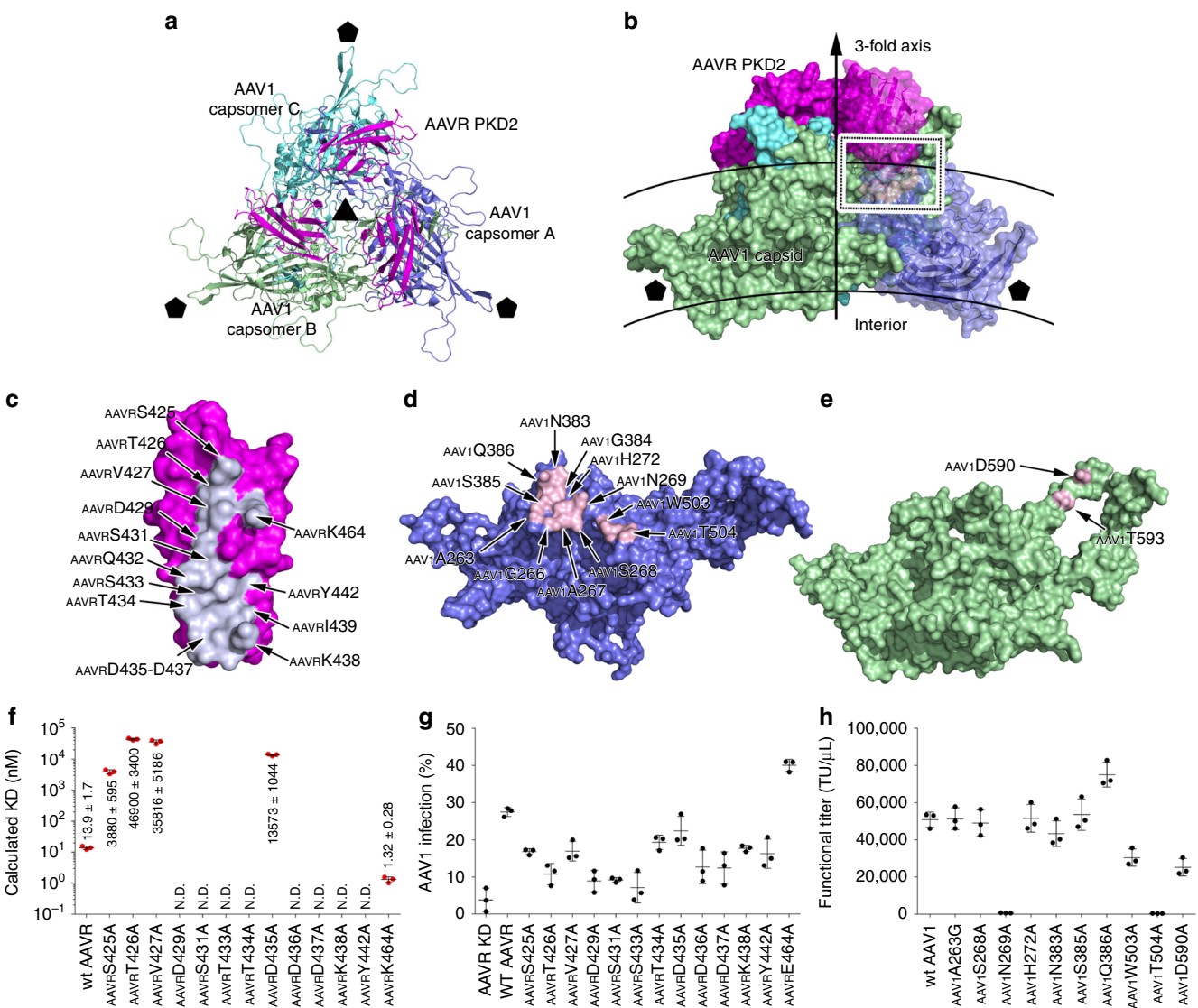

**Fig. 3** AAVR PKD2 binds with AAV1. **a**, **b** The structures of trimeric AAV1 capsomers in complex with AAVR are shown in ribbon representation from a top view (**a**) and as a covered surface from a perpendicular side view (**b**). The rough inner and outer boundaries of the shell are marked with gray arcs. The three AAV1 capsomers are colored blue, green, and cyan, respectively. Bound AAVRs are shown in magenta. The five- and threefold axes are indicated by pentagons and triangles, respectively. The interacting residues on one AAVR and the interacting AAV1 capsomers are colored blue/white and pink in **b**, and the interacting region is framed. The interface amino acid residues in AAVR PKD2 are shown in bluewhite in **c**; the interface residues in AAV2 capsomer A (**d**) and capsomer B (**e**) are labeled and colored pink. See also Supplementary Table 5. **f** A total of 13 AAVR mutants were tested for their ability to bind AAV5 capsid by BIAcore sensorgrams in triplicate experiments. The calculated KD values for the binding of each mutant to AAV1 are summarized. The original curves are shown in Supplementary Fig. 6. **g** The overexpression of AAVR PKD2 mutants in AAVR-silenced HEK293T cells impacted on AAV1 transduction. Cells were transfected with wt AAVR or different AAVR mutants as indicated followed by infection with AAV1-mCherry at an MOI of $5 \times 10^5$ vg cell$^{-1}$. The percentage of mCherry-positive cells are plotted as means ± standard errors ($n = 3$). Expression of wt AAVR and mutant AAVRs was evaluated by immunoblot analysis with β-actin as a control (see Supplementary Fig. 13). **h** AAV1 viral particles bearing capsid mutations at the receptor-binding sites impacted viral transduction. HEK293T cells were infected with serial dilutions of mutant AAV1 starting at an MOI of $5 \times 10^5$ vg cell$^{-1}$. Functional titer refers to the number of virus particles that can infect the cell in every microliter. GFP expression was determined 48 h post-transduction. The functional titers are plotted as means ± standard errors ($n = 3$). Source data are provided as a Source Data file

helix (Fig. 3d, Supplementary Tables 4 and 5). On an adjacent AAV2 capsomer (capsomer B), the residues that participate in PKD2 binding are located in the βGH11-βGH12 loop (residues $_{AAV1}$D590 and $_{AAV1}$T593) (Fig. 3e). The βBC1-βC loop (including residues $_{AAV1}$A263, $_{AAV1}$G266, $_{AAV1}$A267, $_{AAV1}$S268, $_{AAV1}$N269, and $_{AAV1}$H272) and the βEF2-αEF3 loop ($_{AAV1}$N383, $_{AAV1}$G384, $_{AAV1}$S385, and $_{AAV1}$Q386) of capsomer A stack with strands B and C and the BC loop spanning residues $_{AAVR}$S425-$_{AAVR}$Y442 of PKD2. Residues $_{AAV1}$D590′ and $_{AAV1}$T593′ in the βGH11-βGH12 loop from capsomer B also stabilize the virus–receptor interaction

in this region. Moreover, residue $_{AAVR}$K464 in PKD2 strand E makes an additional contact with $_{AAV1}$T593′ in capsomer B.

The binding of PKD2 to the AAV1 capsid resulted in a conformational change at the βGH9-αGH10 loop (Supplementary Fig. 5b), which is the counterpart of the βGH2-βGH3 loop of AAV2 and has conformational movement upon AAVR binding. The counterpart of the AAV2 βBC1-βC loop in AAV1, which underwent a significant shift in the AAVR-bound AAV2 state, shows no obvious structural shift compared with the location of this loop in the unbound AAV1 structure (Supplementary Fig. 5b).

**Impact of interacting residues on AAV1 transduction**. Among the 15 residues of PKD2 from AAVR that participate in the interaction with AAV1, the side chains of $_{AAVR}$S425, $_{AAVR}$T426, $_{AAVR}$V427, $_{AAVR}$D429, $_{AAVR}$S431, $_{AAVR}$S433, $_{AAVR}$T434, $_{AAVR}$D435, $_{AAVR}$D436, $_{AAVR}$D437, $_{AAVR}$K438, $_{AAVR}$Y442, and $_{AAVR}$K464 form non-covalent interactions with AAV1. We individually substituted these residues with alanine residues to address their impact on the AAV1-AAVR interaction in vitro. Wt AAVR binds to AAV1 with a KD value of 13.9 nM (Fig. 3f, Supplementary Fig. 6k–x), which is higher than the binding affinities of AAVR with AAV2 and AAV5. Among 13 mutants, $_{AAVR}$K464A generated a 10-fold increased KD value (1.32 nM), while other mutations resulted in a significant loss of AAV1 binding (Fig. 3f, Supplementary Fig. 6k–x). And as expected, the mutations on PKD1 has no obvious impact on the binding with AAV1 and the mutations on PKD2 did not significantly affect the binding with AAV5, indicating PKD1 or PKD2 are respectively responsible for AAV5 or AAV1 engagement (Supplementary Figs. 9a, b).

We then ectopically expressed these 13 mutants (including $_{AAVR}$S425A, $_{AAVR}$T426A, $_{AAVR}$V427A, $_{AAVR}$D429A, $_{AAVR}$S431A, $_{AAVR}$S433A, $_{AAVR}$T434A, $_{AAVR}$D435A, $_{AAVR}$D436A, $_{AAVR}$D437A, $_{AAVR}$K438A, $_{AAVR}$Y442A, and $_{AAVR}$K464A) in AAVR-silenced cells and evaluated AAV1 transduction in these cells (Fig. 3g). The substitutions at $_{AAVR}$S425, $_{AAVR}$T426, $_{AAVR}$V427, $_{AAVR}$D429, $_{AAVR}$S431, $_{AAVR}$S433, $_{AAVR}$D436, $_{AAVR}$D437, $_{AAVR}$K438, and $_{AAVR}$Y442 attenuated the percentage of AAV1 infection, while the $_{AAVR}$T434A and $_{AAVR}$D435A mutations retained 70% of AAV1 infection. And as expected, the replacement at $_{AAVR}$K464 increased AAV1 transduction (Fig. 3g, Supplementary Fig. 7d). Their effects were consistent with their impacts on the interaction between AAV1 and the recombinant AAVR in vitro. Again, we hypothesized that the $_{AAVR}$K464A mutation might affect the overall folding of AAVR and thus promote AAV1 infection, like the impact of the $_{AAVR}$I349A mutation on AAV5 infection. Again, as expected, the PKD1 mutations or the PKD2 mutations exhibited slight or negligible impacts on the AAV1 or AAV5 transduction, respectively (Supplementary Fig. 9c, d). It is notable that the deletion of PKD2 completely demolishes AAV1 transduction and the deletion of PKD1 decreased 30% AAV1 transduction compared with that in wt cells[11]. We reasoned that the deletion of PKD1, but the single site mutations of PKD1 do not, leads significant change of the overall structure of AAVR, thus resulting in 30% decrease of AAV1 transduction by PKD1 deletion and negligible effects on AAV1 transduction by PKD1 mutations. Nevertheless, these results together demonstrate the minor impact of PKD1 on AAV1 transduction.

We next generated mutated AAV1 viruses containing substituted AAV1 capsid residues at the interface and tested their impacts on viral transduction. The mutation of $_{AAV1}$N269 and $_{AAV1}$T504 completely eliminated AAV1 transduction, and the $_{AAV1}$W503A and $_{AAV1}$D590A mutations attenuated transduction to 30–40% of that observed for wt AAV1 (Fig. 3h, Supplementary Fig. 7c). These results are consistent with the structural observations that these four AAV1 capsid residues form extensive intermolecular contacts with $_{AAVR}$V427, $_{AAVR}$D429, $_{AAVR}$S431, $_{AAVR}$Q432, $_{AAVR}$S433, $_{AAVR}$T434, $_{AAVR}$I439, and $_{AAVR}$Y442 in PKD2 (Fig. 3c–e, Supplementary Table 5). Other mutations including $_{AAV1}$G263A, $_{AAV1}$S268A, $_{AAV1}$H272A, $_{AAV1}$N383A, and $_{AAV1}$S385A showed mild or negligible effects on AAV1 transduction. Consistently, the mutated AAV1 containing $_{AAV1}$N269A, $_{AAV1}$W503A, $_{AAV1}$T504A, and $_{AAV1}$D590A mutations exhibited clear deceased binding with wt AAVR in the virus overlay assays (Supplementary Fig. 8b). Again, we identified an exception; the $_{AAV1}$Q386A mutation increased the functional titer to 1.5-fold that observed for wt AAV1 (from $50803 \pm 4160$ TU μl$^{-1}$

to $75070 \pm 6701$ TU μl$^{-1}$ for wt AAV1 and the mutant, respectively). In the AAV1-AAVR complex, $_{AAV1}$Q386A forms five ideal bonds with $_{AAVR}$D437 to stabilize the AAV1-AAVR interaction, and the mutation of $_{AAVR}$D437 in AAVR attenuated the AAV1-AAVR interaction in the SPR assay and AAV1 transduction. The mechanism for this discrepancy needs further investigation.

**Comparison of the receptor interfaces on the AAV capsids**. Previously structural biology efforts have demonstrated the binding sites of SIA on the AAV1 and AAV5 capsids[13,14]. For AAV1, SIA was observed to bind in a pocket located at the base of capsid protrusions surrounding icosahedral threefold axes[14] (Fig. 4a). For AAV5, two SIA molecules bind at the depression in the center of the icosahedral threefold axis or under the βHI loop close to the fivefold axis, but only the first binding site (site A) plays critical in AAV5 transduction[13] (Fig. 4c). Moreover, previously reported structures of AAV2-HSPG at medium resolution[15] and the structure of AAV-DJ in complex with heparinoid pentasaccharide (an HSPG analog)[16] showed that HSPG binds at the right side of the inner face of the spike on the viral capsid towards the threefold icosahedral axis (Fig. 4b). The PKD2 footprint on both the AAV2 and AAV1 capsids is located at the left side of the inner face of the spike towards the threefold axis and extends to the plateau, although the interacting residues show slight differences (Fig. 4d, e). In sharp contrast, the AAVR-interacting residues on the AAV5 capsid are mainly distributed at the outer face of the spike, and PKD1 from AAVR spans over the canyon region (Fig. 4f). As results, the SIA-binding site is overlapped with PKD2-binding site on the AAV1 capsid, indicating the potential competition between SIA and AAVR, whereas the HSPG or the major SIA-binding region is distant from AAVR-binding site on the AAV2 or AAV5 capsids.

There are nine variable loop regions (VR-I to VR-IX) on the surface of the AAV capsid with the most variability between different AAV genotypes[17]. Our previous work has shown that AAVR-interacting residues on the AAV2 capsid are distributed in VR-I, VR-III, VR-VI, and VR-VIII[12]. In AAV1, the residues that interact with PKD2 are in VR-I, VR-III, and VR-VIII, while the residues in AAV5 that engage PKD1 are located in VR-II, VR-VII, and VR-IX (Supplementary Fig. 10, Supplementary Tables 6 and 7).

Three loop regions of the AAV2 capsid surround the bound HSPG (Fig. 4g–i). The first region maps to the βGH6-αGH6′ loop in the AAV2 capsid, which corresponds to the βGH4-αGH5 loop in AAV1 and the βGH7-αGH8 loop in AAV5 (Fig. 4g). In this region, the conformation of the βGH7-αGH8 loop in AAV5 is more extended than the βGH6-αGH6′ loop in AAV2 and the βGH4-αGH5 loop in AAV1. In the second and third regions, the loops from all three AAV serotypes display high structural similarity (Fig. 4h, i). Sequence alignments showed that although the third region has a relatively high sequence identity, the first and the second regions are less conserved (Supplementary Fig. 10). Notably, in the third region, two arginine residues in AAV2, $_{AAV2}$R585 and $_{AAV2}$R588, directly bind with HSPG or its analogs, but the corresponding residues in AAV1 and AAV5 are serine, threonine, or asparagine residues, which do not have long side chains with positive charges (Supplementary Fig. 10). This may explain why HSPG binding plays a dominant role in AAV2 entry, but not in AAV1/4/5/6 entry[18–21].

The SIA-binding pocket on the AAV1 capsid is located at the base of capsid protrusions surrounding icosahedral threefold axes[14] (Fig. 4j). Although the major structural parts constituting this SIA-binding pocket on the AAV1/2/5 capsids show high structural conservation, the regions corresponding to the βBC2-βC loop of

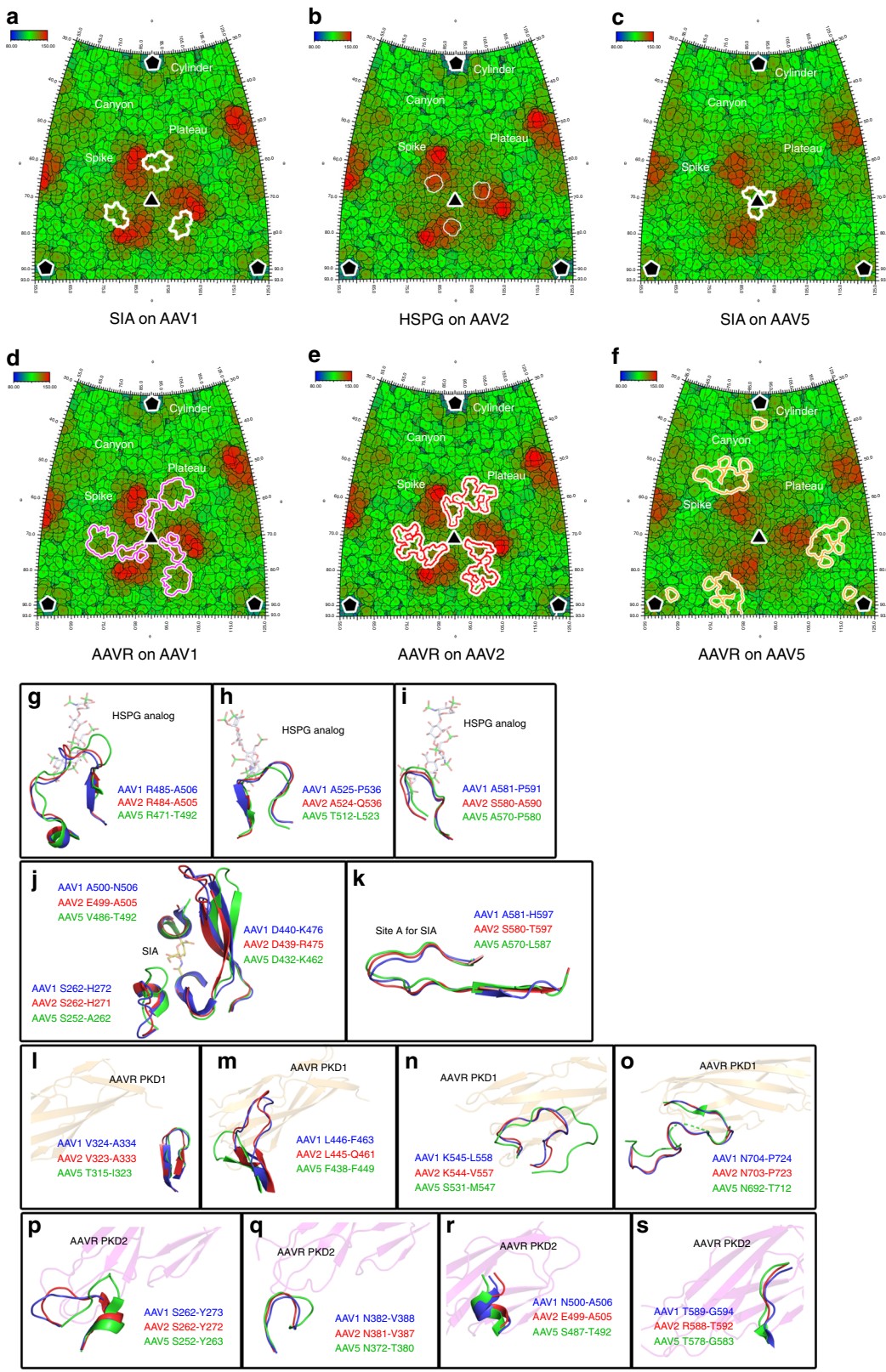

AAV1 have large conformational discrepancy (Fig. 4j). Particularly, this part on the AAV5 capsid has a remarkable larger loop region shadowing the SIA binding pocket. There are two SIA-binding sites found on the AAV5 capsid, but only site A play essential role for AAV5 transduction[13]. Notably, the structures of site A on the AAV5 capsid and the corresponding regions on the AAV1 and AAV2 capsids display high similarities (Fig. 4k).

PKD1 interacts with the AAV5 capsid mainly via four loop regions including the βD-βDE1 loop (region 1), the βGH4-βGH5 loop (region 2), the βGH13-βGH14 loop (region 3), and the βI-βI1 loop (region 4) (Fig. 4l–o). In regions 2 and 4, the loops from AAV1, AAV2, and AAV5 adopt similar conformations (Fig. 4l, o). In contrast, AAV5 has a remarkably shorter loop in region 2 and a longer loop in region 3 than those in AAV1 and

**Fig. 4** Comparison of the binding between receptors and AAV capsids. **a–f** Roadmap depictions of the AAV1, AAV2, and AAV5 icosahedral surfaces projected onto a plane and showing an area larger than one icosahedral face (outlined as a triangle with the three- and fivefold vertices marked with triangles and pentagons, respectively). Two angles ($\theta$, $\phi$) define a vector and thus a location on the icosahedral surface. The roadmaps are radially depth cued, as shown by the key, from blue (radius = 80 Å) to red (radius = 150 Å). The footprints of the HSPG analog and SIA, or AAVR on different AAV capsids are outlined by white, purple, red, and orange lines. **g–i** Three structural elements of the AAV2 capsid surrounding the bound HSPG analog are shown as cartoon diagrams and colored red, whereas their counterparts in AAV1 and AAV5 are colored blue and green, respectively. The bound HSPG analog is displayed as semi-transparent sticks. **j** The structural elements of the AAV1 capsid surrounding the bound SIA are shown as cartoon diagrams and colored blue, whereas their counterparts in AAV2 and AAV5 are colored red and green, respectively. The bound SIA is displayed as semi-transparent sticks. **k** The structural elements of the AAV5 capsid forming the site A to bind SIA are shown as cartoon diagrams and colored green, whereas their counterparts in AAV1 and AAV2 are colored blue and red, respectively. The bound SIA is displayed as semi-transparent sticks. **l–o** Four structural elements of the AAV5 capsid interacting with PKD1 and their counterparts in the AAV1 and AAV2 capsids are shown as cartoon diagrams with the same color scheme above. The bound PKD1 is represented as a semi-transparent cartoon. **p–s** Four structural elements of the AAV1 capsid interacting with PKD2 and their counterparts in AAV2 and AAV5 are shown as cartoon diagrams with the same color scheme above. The bound PKD2 is represented as a semi-transparent cartoon

AAV2 (Fig. 4m, n). Consistent with this, regions 1 and 4 in all three AAV serotypes are highly sequence conserved, but region 3 is less similar, indicating that both the conformation and amino acid sequence of region 3 are essential for AAV5 recognition of PKD1.

PKD2 interacts similarly with AAV1 and AAV2 mainly through the βBC2-βC loop (region 1), the βEF2-αEF3 loop (region 2), the βGH4-αGH5 loop and αGH5 (region 3), and the βGH11-βGH12 loop (region 4) (Fig. 4p–s). Regions 2, 3, and 4 exhibit high structural and sequence homology. In contrast, region 1 (VR-I) is less conserved in AAV1/2 than in AAV5. Consistent with this, the conformation of region 1 in AAV5 is distinct, indicating that this region could determine the interaction of AAV1/2 with PKD2. Although PKD2 binds to AAV1 and AAV2 in a similar manner, three major interacting residues are different. For AAV2, the $_{AAV2}$R471, $_{AAV2}$E499, and $_{AAV2}$K507 residues interact with PKD2. However, their counterparts in AAV1, i.e., $_{AAV1}$S472, $_{AAV1}$N500, and $_{AAV1}$K508, are distant from the interface with PKD2 (Supplementary Fig. 11). In AAV2, the amino group of the side chain of $_{AAV2}$R471 forms a 3-Å hydrogen bond with the side chain atom of $_{AAVR}$Q432. In contrast, the counterpart of $_{AAV2}$R471 in AAV1 is a serine residue ($_{AAV1}$S472); thus, the interaction with $_{AAVR}$Q432 is absent due to the short serine side chain. Similarly, the side chain carbon atom of $_{AAV2}$E499 faces $_{AAVR}$P414, forming a van der Waals contact to stabilize AAV1-AAVR binding; however, the shorter side chain of $_{AAV1}$N500 prevents this interaction. Although there is a lysine residue in position 508 of AAV1 and position 507 of AAV2, the conformation of these lysine side chains in AAV1 and AAV2 are slightly shifted, thus preventing the interaction between this lysine residue and $_{AAVR}$I462 in the AAV1-AAVR pair.

**The impact of the AAVR glycosylation site in PKD2 on AAV5 transduction.** AAVR is a glycosylated protein with five N-linked asparagine glycosylation sites[11]. A previous study showed that only the mutation of N525 to alanine to prevent N-linked glycosylation mildly affected AAV2 transduction, whereas the other mutations, $_{AAVR}$N395A, $_{AAVR}$N472A, $_{AAVR}$N487A, and $_{AAVR}$N492A, did not appreciably affect the transduction efficiency[11]. The AAV1-AAVR and AAV2-AAVR complex structures clearly show that $_{AAVR}$N472, $_{AAVR}$N487, and $_{AAVR}$N492 in PKD2 are located on the opposite side of the AAV1/AAV2-AAVR interface and that their side chains are distant and point outward away from the viral capsids (Supplementary Fig. 12a). In the AAV5-AAVR complex, although $_{AAVR}$N395 is not directly involved in an interaction with the AAV5 capsid, its side chain points towards the AAV5 capsid at the closest distance of 6.5 Å (Supplementary Fig. 12b). As we used bacterially expressed AAVR in this structural study, the extended glycosylation

moieties of $_{AAVR}$N395 likely contact the AAV5 capsid. As expected, when we mutated N395 to an alanine residue, AAV5 transduction was attenuated (Fig. 2g). Because PKD2 does not directly bind AAV2, it is not surprising that the mutation of $_{AAVR}$N395 did not affect AAV2 transduction in a previous study[11]. It should be noted that $_{AAVR}$N525 is located in PKD3, and we cannot observe its direct interaction with the AAVs in all AAV-AAVR structures.

## Discussion

Virus–receptor recognition is an important prerequisite for viral infection. For nonenveloped viruses, one kind of receptor usually engages multiple virus genotypes/serotypes within one species or multiple viral species with a closed evolutionary relationship, with a conserved rule. For example, picornaviruses, a family of nonenveloped viruses, engage receptors with Ig-like domains through conserved mechanisms. Intracellular adhesion molecule-1 (ICAM-1 or CD54) attaches to human rhinoviruses 14 and 16 (HRV14 and HRV16) and coxsackievirus A21 (CVA21), poliovirus receptor (PVR or CD155) binds with poliovirus, and coxsackievirus adenovirus receptor (CAR) makes contact with several coxsackievirus B serotypes (CVB1-CVB6) via the similar interactions between the D1 Ig-like domain of the receptors and the canyon region of the picornavirus capsids[22–27]. AAVR is a canonical Ig-like protein with five Ig-like PKD domains. The structures of AAVR in complex with AAV1, AAV2, and AAV5 demonstrate that PDK1 employs AAV5 attachment, whereas PKD2 binds directly with AAV1 and AAV2, which is the first interesting example of a single receptor attached to multiple viral strains with divergent rules.

Previous results showed that multiple AAV serotypes (including AAV1 but not AAV2/AAV5) required a combination of PKD1 and PKD2 for their efficient transduction[10,11]. However, PKD1, which acted together with PKD2 for AAV1 entry in a functional assay[11], could not be found in the AAV1-AAVR complex, which is consistent with the previous reported result that AAV1 binds only to PKD2 in a virus overlay assay[11]. Moreover, although AAV5 binds PKD1 and AAV1/2 binds PKD2, the PKD1- and PKD2-interacting regions on the AAV1/2 and AAV5 capsids have limited sequence variation. It is conceivable that the combination of the PKD1- and PKD2-interacting components in one AAV capsid may result a recombinant AAV vector to contact both PKD1 and PKD2, thus increasing its binding affinity with AAVR. Furthermore, the mutations of a few individual sites on the AAV5-AAVR and AAV1-AAVR interfaces increased but not attenuate viral infectivity. These sites are also potential sites to optimize AAV transduction efficacy in therapeutic applications.

The capsids of AAV serotypes determine the properties of tissue tropism and antigenic properties[28]. Among the currently identified 13 distinct AAV serotypes, AAV2 is the traditionally used vector with board tissue tropisms[3]. Recent progress demonstrated that AAV1 is far more efficient in transducing muscle than AAV2 (ref. [29]). We reason the higher transduction efficiency of AAV1 may be caused by its remarkable higher AAVR binding affinity (13.9 nM) than that of AAV2 (57 nM). Notably, the monospecific antibodies to AAV1 have not been detected in human serum[30]. These properties lead the potential clinical application of AAV1 in patients who develop anti-AAV2-neutralizing antibodies due to a naturally acquired infection or previous treatment with AAV2 vectors[29,30]. AAV5 is distinct among the dependent parvoviruses, since it was originally isolated from a patient sample instead of from laboratory stocks of adenovirus and has unique sequence and biochemical features[31]. It was shown that AAV2 has 6- to 32-fold higher transduction efficiencies than AAV5 in cos, 293, HeLa, IB3, and MCF7 cells, but both viruses exhibited poor transduction efficiencies in NIH 3T3, skbr3, and t-47D cell lines[31]. Moreover, AAV5 transduction is not sensitive to heparin, which inhibits AAV2 (refs. [6–8]). These distinct features could be related with the binding of virus particles with AAVR, or in the combination of AAVR and different glycan moieties.

In summary, combined with the results from our previous work[32], the atomic structures of AAVR in complex with multiple AAV serotypes further the understanding of the divergent engagements between multiple AAV serotypes with AAVR and provide information to facilitate the design and optimization of AAV vectors for gene therapy. Our results also present an example of one receptor engaging multiple genotypes/serotypes with divergent rules in virology.

## Methods

**Cell culture**. Human embryonic kidney 293T (HEK293T) cells (Thermo Fisher, USA) were maintained in Dulbecco's modified Eagle's Medium (DMEM) supplemented with 10% fetal bovine serum (FBS) and 1% penicillin–streptomycin in an incubator at 37 °C with 5% $CO_2$.

**Virus production and purification**. Triple-plasmid transfection using polyethylenimine reagent (PEIMAX) (No. 24765; Polysciences, USA) was carried out to produce recombinant AAV1 and AAV5 according to a previously reported procedure with modifications[12]. The plasmids pAAV1-GFP (or pAAV1-mCherry) or pAAV5-GFP (or pAAV5-mCherry); pRepCap with AAV1 or AAV5 encoding the Rep and Cap proteins or mutant pRepCap plasmids carrying various capsid mutations; and pHelper plasmids were cotransfected into HEK293T cells. For the large-scale production of AAVs, cells were seeded in 150 mm dishes at a density of $1 \times 10^7$ cells per dish 24 h prior to transfection and transfected with 12 μg pHelper plasmid, 8 μg AAV1 or AAV5 pRepCap plasmids, and 5 μg pAAV1 or pAAV5 plasmids at 70% confluency. For the small-scale production of AAVs, cells were seeded in 24-well plates at a density of $1.5 \times 10^5$ cells per well 24 h prior to transfection and transfected with 300 ng pHelper plasmid, 190 ng AAV1 or AAV5 pRepCap plasmids, and 140 ng pAAV1 or pAAV5 plasmids at 70% confluency. At 72 h post-transfection, cells were harvested by centrifugation at $500 \times g$ at 4 °C for 5 min. The pellet was collected and resuspended in lysis buffer containing 50 mM Tris-HCl, pH 8.0, 150 mM NaCl. The suspension was subjected to three freeze–thaw cycles in dry ice/ethanol and a 37 °C water bath. Then, benzonase (100 units per ml) and 0.5% sodium deoxycholate were added, and the cell suspension was incubated for 1 h at 37 °C. Following centrifugation at $5000 \times g$ for 20 min at 4 °C, the supernatant containing the AAV crude lysate was collected. AAV genome copy titers were determined by real-time quantitative PCR (qPCR) using primers specific for the GFP or mCherry gene sequences. The primers used were as follows: qpcr-GFP-F, TCTTCAAGTCCGCCATGCC; qpcr-GFP-R, TGTCGCCC CTCGAACTTCAC; qpcr-mCherry-F, AGATCAAGCAGAGGCTGAAGCTGA; and qpcr-mCherry-R, ACTGTTCCACGATGGTGTAGTCCT.

The crude lysate was diluted with a 10 mM Tris-HCl (pH 8.0) buffer to a final volume of 10 ml and then bottom loaded on a discontinuous gradient of 15%, 25%, 40%, and 60% iodixanol in a 39 ml ultracentrifuge tube (Quick-Seal 342414, Beckman, USA). After ultracentrifugation with $350,000 \times g$ at 18 °C for 1 h, 3 ml fractions in the 40% lower layer and 0.5 ml of the 60% upper layer were collected. The viral titers in each fraction were determined by qPCR. The fractions with the highest titers were desalted using a 100 kDa cutoff ultrafiltration tube (15 ml;

Millipore, USA), and the buffer was changed to PBS. Purified AAV1 and AAV5 were stored at −80 °C until use.

**Purification of wt and mutated AAVR proteins**. Recombinant AAVR was expressed and purified as previously reported[12]. cDNAs encoding the wt AAVR PKD1–5 domains with a C-terminal His-tag in a pET28a vector were transformed into *Escherichia coli* BL21 (DE3). *E. coli* cells harboring the recombinant plasmids were cultured in Luria-Bertani medium containing 50 μg ml−1 kanamycin at 37 °C. When the OD$_{600}$ of the culture reached 0.6, protein expression was induced by the addition of isopropyl β-D-thiogalactoside at a final concentration of 0.5 mM, followed by another 6 h of cell culture. The cells were harvested by centrifugation at 4500 r.p.m. for 12 min. The pellet was resuspended in 1 × PBS, pH 7.4, with 100 μM protease inhibitors and then homogenized by sonication. The fusion protein was then isolated from its crude lysate by nickel affinity chromatography (Qiagen, Holland) and dialyzed into buffer containing 20 mM HEPES, pH 7.4, and 20 mM NaCl. The protein was further purified by anion exchange chromatography using a Resource Q column (GE Healthcare, USA) and stored at −80 °C. Mutated AAVR constructs were generated using the Fast Mutagenesis System (Transgene, China), and the mutated proteins were purified as described above. The purified proteins were concentrated to 1 mg ml−1 for storage at −80 °C until use.

**Sample preparation and cryo-EM data collection**. AAV1 or AAV5 particles and purified wt AAVR were mixed at a molar ratio of 1:120 (AAV:AAVR) at 4 °C for 6 h. An aliquot of 3 μl of each mixture was loaded onto a glow-discharged, carbon-coated copper grid (GIG, Au 1.2/1.3 200 mesh; Lantuo, China) bearing an ultrathin layer of carbon. The grid was then blotted for 4 s with a blot force of 0 in 100% relative humidity and plunge-frozen in liquid ethane using a Vitrobot Mark IV (FEI, USA). Cryo-EM data were collected with a 200 kV Arctica D683 electron microscope (FEI, USA) and a Falcon II direct electron detector (FEI, USA). A series of micrographs were collected as movies (19 frames, 1.2 s) and recorded with −2.2 to −0.5 μm defocus at a calibrated magnification of ×110,000, resulting in a pixel size of 0.93 Å per pixel. Statistics for data collection and refinement are summarized in Supplementary Table 1.

**Image processing and three-dimensional reconstruction**. Similar image processing procedures were employed for all data sets. Individual frames from each micrograph movie were aligned and averaged using MotionCor2 (ref. [33]) to produce drift-corrected images. Particles were picked and selected in RELION 2.1 (ref. [34]), and the contrast transfer function (CTF) parameters were estimated using CTFFIND4 (ref. [35]). Subsequent steps for particle picking and 2D and 3D classification were performed with RELION 2.1 (ref. [34]). The final selected particles were reconstructed with THUNDER[36]. For all reconstructions, the final resolution was assessed using the gold-standard FSC criterion (FSC = 0.143) with RELION 2.1 (ref. [34]).

**Model building and refinement**. To solve the structure of AAV1, the X-ray crystal structure of AAV1 (PDB code: 5EGC)[14] was manually placed and rigid body fitted into the cryo-EM density map with UCSF Chimera[37]. To solve the AAV1-AAVR complex, the PKD2 domain structure from the AAV2-AAVR structure (PDB: 6IHB) was manually aligned with cryo-EM density corresponding to the bound receptors. To solve the structure of AAV5, the previously reported structure of AAV5 (PDB code: 3NTT)[38] was manually placed and rigid body fitted into the density for AAV5 alone with UCSF Chimera[37]. To solve the AAV5-AAVR complex, the structure of the PKD domain (residues 329–428) from human KIAA0319 (PDB: 2E7M) was manually aligned into the cryo-EM density corresponding to the bound receptors. The fit was further improved with real-space refinement using Phenix[39]. Manual model building was performed using Coot[40] in combination with real-space refinement with Phenix to replace the corresponding amino acids in the model with those from PKD1 of AAVR. The density maps were kept constant during the entire fitting process, and only the atomic coordinates were subjected to refinement. The data validation statistics shown in Supplementary Table 1 were reported by MolProbity using the integrated function within the Phenix statistics module[39].

**Surface plasmon resonance (SPR)**. SPR analyses were carried out using a Biacore T200 (GE Healthcare, USA) with a flow rate of 30 μl min−1 at 25 °C in PBS buffer. AAV1 or AAV5 particles suspended in sodium acetate buffer (pH 4.5) were immobilized on a CM5 sensor chip by amide coupling. Different concentrations of the recombinant wt or mutated AAVR proteins flowed over the chip. The binding affinity was determined by and curves were generated by BIAEvaluation software (GE Healthcare, USA).

**Production of the mutated the AAVs**. A ClonExpress Ultra One Step Cloning Kit (Vazyme, China) was utilized to introduce mutations in selected AAV1 or AAV5 capsid residues at the AAVR-binding interface. pRepCap plasmids carrying the wt Rep and Cap protein coding sequences of AAV1 or AAV5 were used to generate the capsid mutants. The obtained mutant clones were confirmed by Sanger DNA sequencing of the full-length ORFs. The pAAV1-GFP or pAAV5-GFP plasmids, the mutated pRepCap plasmids with various AAV1 or AAV5 capsid mutations,

and the pHelper plasmid were cotransfected into HEK293T cells to produce the recombinant viruses. All primer sequences are listed in Supplementary Table 8.

**AAV infection assays.** Recombinant AAV vectors carrying a GFP expression cassette were used to evaluate viral transduction efficiency. At 24 h prior infection, HEK293T cells were seeded in 96-well plates at a density of $2.5 \times 10^4$ cells per well. Cells were infected with purified AAV1-GFP or AAV5-GFP in triplicate with 10-fold serial dilutions starting at a multiplicity of infection (MOI) of $5 \times 10^5$ or $3 \times 10^6$ viral genome (vg) per cell. At 48 h post-infection, GFP expression levels were measured by flow cytometry using a CytoFLEX LX analyzer (Beckman, USA). Wells with fewer than 40% GFP-positive cells were chosen to calculate the transduction titer (TU $\mu l^{-1}$) using the following formula: (total transduced cell number × %GFP)/(virus volume in $\mu l$).

**Generation of AAVR-silencing cell lines.** For lentivirus production, 24 h prior to transfection, cells were seeded into six-well plates at a density of $5 \times 10^5$ cells per well and transfected at 70% confluency. Cells were cotransfected with the shRNA expression plasmid pHS-GFP-2A-Puro-shRNA-3 (ref. [12]), the packaging plasmid pCMV-dR8.9 (Addgene, USA), and the envelope plasmid pCMV-VSV-G (Addgene, USA) using PEIMAX (Polysciences, USA). The medium was exchanged with fresh lentivirus harvesting medium 8 h post-transfection. Medium containing lentiviral particles was harvested 48 h post-transfection and centrifuged to remove cellular debris. A total of 500 $\mu l$ of the supernatant and 8 $\mu g\ ml^{-1}$ Polybrene was added to a 24-well plate seeded with $2 \times 10^5$ cells per well 24 h prior to transduction. At 48 h post-transduction, 1 $\mu g\ ml^{-1}$ puromycin (Thermo Fisher, USA) was added and maintained for 7 days.

**Immunoblot analysis.** Cells were lysed with RIPA buffer (50 mM Tris-HCl, pH 8.0, 150 mM NaCl, 1% NP-40, 0.1% SDS and 0.5% sodium deoxycholate) on ice for 30 min and centrifuged at $14,000 \times g$ for 20 min to remove the cellular debris. Protein samples were prepared by mixing the cell lysate with $5 \times$ loading buffer and boiling for 5 min at 95 °C. Proteins were separated on a SurePAGE™ Bis-Tris gel and transferred onto polyvinylidene fluoride membranes (GE Healthcare, USA). Primary antibodies were used at a dilution of 1:800 (anti-KIAA0319 antibody, ab118923, Abcam, USA) or 1:1000 (anti-β-actin antibody, 66009-1-Ig, Proteintech, USA), and horseradish peroxidase (HRP)-conjugated secondary antibodies (anti-mouse or anti-rabbit in TBST, A16066, 31460, Invitrogen, USA) were used at a dilution of 1:10000. Signals were detected on a ChemiDoc™ MP Imaging System (Bio-Rad, USA).

**Ectopic expression of AAVR mutant and reporter assays.** AAVR-silenced HEK293T cells were used for AAVR mutant transfection and infected with wt AAVs for reporter assays. Mutants in the PKD1 or PKD2 domains were introduced into the eukaryotic expression vector pcDNA3.1(+)-AAVR by the Fast mutagenesis system (TransGen Biotech, China). At 24 h prior to transfection, cells were seeded into 96-well or 12-well plates at a density of $2.5 \times 10^4$ cells per well or $3 \times 10^5$ cells per well and transfected at 70% confluency using PEIMAX. A total of 100 ng plasmids was used for each well of the 96-well plate, or a total of 1000 ng plasmids was used for each well of the 12-well plate. At 8 h post-transfection, the medium was changed to AAV1-mCherry- or AAV5-mCherry-containing medium at a MOI of $5 \times 10^5$ or $3 \times 10^6$ vg vg cell$^{-1}$. At 48 h post-transduction, mCherry-positive cells were detected by flow cytometry by using a CytoFLEX LX analyzer (Beckman, USA). All primer sequences are listed in Supplementary Table 9.

**Virus overlay assay.** The virus overlay assay was performed as previously described[11,41]. In details, 6 $\mu g$ purified wt AAVR were separated on a Bis-Tris gel and transferred onto polyvinylidene fluoride membranes (GE Healthcare, USA). The membrane was sequentially incubated with TBST blocking buffer (Tris-buffered saline—0.1% Tween 20 with 10% nonfat milk), AAV capsid mutants crude lysate at ~$2.5 \times 10^{11}$ vg ml$^{-1}$ in TBST–2% nonfat milk, an anti-AAV VP1/VP2/VP3 mouse monoclonal, B1 (No. 65158; Progen, German) at a 1:200 dilution in TBST–2% NFT, and horseradish peroxidase (HRP)-conjugated secondary antibodies (anti-mouse, A16066; Invitrogen, USA) at a 1:10,000 dilution in TBST, Signals were detected on a ChemiDoc™ MP Imaging System (Bio-Rad, USA).

**Reporting summary.** Further information on research design is available in the Nature Research Reporting Summary linked to this article.

## Data availability
The cryo-EM density maps and the structures were deposited into the Electron Microscopy Data Bank (EMDB) and Protein Data Bank (PDB) with the following accession numbers: AAV1 alone, EMD-9795, PDB 6JCR; AAV1-AAVR, EMD-9794, PDB 6JCQ; AAV5 alone, EMD-9797, PDB 6JCT; and AAV5-AAVR, EMD-9796, PDB 6JCS. The source data underlying Figs. 2f–2h, 3f–3h, and Supplementary Figure 9 are provided as a Source Data file. All other data supporting the findings of this study are available from the corresponding authors upon request.

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

## Acknowledgements

We thank the Computing and Cryo-EM Platforms of Tsinghua University, Branch of the National Center for Protein Sciences (Beijing) for providing facilities. We thank Dr. Lingpeng Cheng for his help in data collection. This work was supported by the National Program on Key Research Project of China (2017YFC0840300 and 2018YFA0507200), and the National Natural Science Foundation of China (grants no. 81322023, 31770309, 81372284, 81520108019, and 31370733).

## Author contributions

Z.L., W.D., and Zihe Rao conceived the project. Z.L. and W.D. designed the experiments. R.Z., G.X., L.C., Z.S., M.H., Y.H., M.C., M.H., H.L., L.Q., Z.Y., and Zipei Rao. performed experiments. R.Z., G.X., L.C., X.L., Y.S., S.L., W.D., and Z.L. analyzed the data. Z.L. wrote the manuscript. All authors discussed the experiments, read, and approved the manuscript.

## Additional information

**Competing interests:** The authors declare no competing interests.

