## [Peer Review File · Nature Communications]

Reviewers' Comments:

Reviewer #1:

Remarks to the Author:

The manuscript by Zhang et al., identifies the interactions of Adeno associated virus serotypes 1 and 5 with cellular receptor, AAVR, using cryo-EM and 3D reconstruction. The structures were resolved to 3.30 Å for AAV1-AAVR and 3.18 Å for AAV5-AAVR. AAVR is a glycoprotein, which contains a cellular membrane spanning domain and 5 polycystic kidney disease domains (PKD1-5). AAV1 specifically interacts with PKD2, whilst AAV5 interacts with PKD1. This group previously published the structure of AAV2 complexed with AAVR, also using cryo-EM and showed that AAV2 binds to PKD2. Mutagenesis followed by transduction assays confirmed the AAV and AAVR binding interface, and showed the importance of some of these residues in viral transduction. Using surface plasmon resonance, the binding affinities of variant AAVRs were assessed against AAV1 and AAV5, with variants resulting in decreased or higher KDs. Mutated AAVR was also ectopically expressed in AAVR knockdown HEK293T cells, to observe the effect on viral transduction efficiency, with variant KD correlating with viral transduction. Capsid variants had a spectrum of phenotypes from no effect to an increase in transduction. This study represents the second report of the ubiquitously acting AAVR complexed with AAV capsids. However, while there is no major concerns with the context of this manuscript, it does not rise to the level for publication in Nature Communications following the publication of the AAV2-AAVR complex structure in Nature Microbiology (Zhang et al, Nature Microbiology 4, pages675–682 (2019)). This current manuscript reads like a continuation of this previous publication.

Minor Comments

Main text

Please run a spell check to correct typos (e.g. AA1)

Lines 53-59 - Include reference to Dependoparvovirus group definitions in Cotmore et al - https://talk.ictvonline.org/ictv-reports/ictv_online_report/ssdna-viruses/w/parvoviridae/1043/genus-dependoparvovirus

Results

Line 120 - PKD2 also spans the 2/5 fold wall of the capsid

Line 135, 198 – these are non-covalent interactions, or weak hydrophobic interaction

Line 141-143 – The AAV5 structure citation does not define secondary structure in this manner, cite the supplemental definition.

Line 162-163 - Why is AAV2 used as control? Could be typo? Need AAV5. If this is AAV5, comparison with AAVR-AAV2 is not conclusive since AAV2 binds PKD2.

Line 233 - Intermolecular bonds needs to be change to non-covalent interactions

Line 247 - Correct 3-G reference to residue K464 from K446.

Line 261 - AAV1 N383A not G

Line 265 - Mutant and AAV1 needs to be swapped in sentence

Line 269 - not sure they have evidence for conformational alteration

Comparison of the receptor interface on the AAV capsids- should keep nomenclature consistent, text is repeated from previous AAV2 paper mentioned above. This section is not really clear, and is most apparent in line 289-311

Line 282-283 - appears in previous AAV2-AAVR paper with a one word changed. Please rephrase

Line 287 - Missing R for VR

Line 298 - Typo with residue numbers should be 585 and 588

Line 301-302- Why use HSPG instead of sialic acid since both AAV1 and AAV 5 bind sialic acid?

Compare the glycan binding pockets for sialic acid with AAV1 and AAV5 - the information for these are

published (fig 4). Huang et al, J Virol. 2016 May 12;90(11):5219-5230; Afione et al, J Virol. 2015 Feb;89(3):1660-72). Does the AAVR pocket overlap in anyway?
Line 326 - P414 instead of P432 for supplementary figure 9
line 347-348 typo- AAV5 instead of AAV2?

Methods

Cell culture- no antibiotics mentioned
Line 397 - need a product number for PEIMAX
Lines 465-472 - The references, 27, 28 and 29 should be updated to the programs actually used
Lines 475 - Why use AAV2 crystal structure instead of AAV1 – see RCSB pdb No. 3NG9.
Line 548 - Why overexpression (just expression)
Line 554 - 1000 ng instead of 100ng

Supplemental

Supplemental figure 2 does not include residue 355
Supplemental fig 4; the body of the texts keeps mentioning PKD2 but it should be PKD1 since referring to AAV5 binding.

Reviewer #2:

Remarks to the Author:

The manuscript by Lou and colleagues is focused on mapping divergent interactions of AAVR with different AAV serotypes, specifically AAV1/2/5 through cryoEM. The study clearly demonstrates that AAV5 interacts with AAVR through PKD1 and AAV1 does so by interacting with PKD2. The striking differences underlying the mechanics of how AAVR is recognized by two distinct AAVs using different surface domains/footprints for docking are a particularly novel finding in the current study. The study is carefully executed, technically sound, comprehensive in utilizing biophysical analysis as well as biological experiments (infectivity/transduction) and the manuscript is well written. Overall, the findings are novel, provide new insight into AAV biology - particularly, the study breaks new ground on how AAVR interactions might dictate AAV cell entry and transduction.

Some concerns are outlined below.

First, are the mutations affecting AAV infection in a serotype specific manner or a global fashion? i.e., are AAVR PKD1/2 mutants able to rescue the non-cognate serotype? Specifically, do PKD1 mutants rescue AAV1 infection or PKD2 mutants rescue AAV5 infection?

Second, what is the impact of AAV capsid mutations on binding to wt AAVR or their respective PKD domains? Only transduction data is shown (the term "Functional titer" in Figures 2H/3H needs to be better defined).

Third, the results should be discussed against the backdrop of the glycan binding footprint on AAV1 and AAV5, both of which have been shown to bind N-linked sialic acid. Further, it is unclear how interaction of different AAVs with the same AAVR (albeit through different sites) can account for the divergent tissue tropisms observed. This should be highlighted as well.

Reviewer #3:

Remarks to the Author:

This study is investigating the structural basis of the interaction between AAVR and different AAV strains. The authors determined four high resolution structures and showed that AAV1 binds to the second domain of AAVR (PKD2, similar to AAV2) whereas AAV5 binds to PKD1. Sequence alignments and structural comparisons were used to interpret the results.

The topic is of general interests and the findings are very inspiring. The manuscript is clearly written, and the figures are professionally prepared.

Two important tasks behind the story are: 1. To show that indeed PKD1 is observed in the map of AAV5-AAVR; 2. To experimentally determine the roles of different amino acids that cause these distinct binding patterns. The authors have done good jobs addressing these issues.

Here are a few specific comments:

1. Since the cryo-EM structures are the foundations of this manuscript, would it be possible to release the maps, either through the EMDB deposition system or private links?
2. A lot of data are presented to show that AAV5 indeed binds to PDK1, but not much is discussed to depth. For example, what might be the biological advantage behind this phenomenon? Is there a particular reason why different AAVs evolved to recognize different regions on the same protein?
3. In both cases of AAV1 and AAV5 complexes, a protein containing all five domains was used. Is there any trace of density coming from other domains (PKD1 and 3 for AAV1, PKD 2 for AAV5)? If not, why is the molecule so flexible?
4. The mutagenesis performed for AAV1-PKD2 and AAV5-PKD1 are mutually exclusive. Since a major point is being made comparing these two complexes, is it possible to see how mutations on PKD1 may affect the binding between AAVR and AAV1, and how mutations on PKD2 affect the binding between AAVR and AAV5? Results on a couple of mutants would be good enough.
5. Mutations on AAVRT350 and AAVRR353 abolished binding to AAV5, but the infectivity was not much influenced, why?
6. Are the clashing scores calculated for the entire capsids? Was icosahedral symmetry applied in the model refinements? What are the correlation coefficients between the maps and the models (such value is usually reported by phenix real space refinement)?
7. It would be helpful to label residues that are conserved between AAV1, 2 and 5 in supplementary tables 3 and 5.
8. Two potential typos: In line 230 and 749, should it be "PKD2" instead of "PKD1"? The caption of sup figure 6 says "supplementary figure 4".

Response to Reviewer #1

The manuscript by Zhang et al., identifies the interactions of Adeno associated virus serotypes 1 and 5 with cellular receptor, AAVR, using cryo-EM and 3D reconstruction. The structures were resolved to 3.30 Å for AAV1-AAVR and 3.18 Å for AAV5-AAVR. AAVR is a glycoprotein, which contains a cellular membrane spanning domain and 5 polycystic kidney disease domains (PKD1-5). AAV1 specifically interacts with PKD2, whilst AAV5 interacts with PKD1. This group previously published the structure of AAV2 complexed with AAVR, also using cryo-EM and showed that AAV2 binds to PKD2. Mutagenesis followed by transduction assays confirmed the AAV and AAVR binding interface, and showed the importance of some of these residues in viral transduction. Using surface plasmon resonance, the binding affinities of variant AAVRs were assessed against AAV1 and AAV5, with variants resulting in decreased or higher KDs. Mutated AAVR was also ectopically expressed in AAVR knockdown HEK293T cells, to observe the effect on viral transduction efficiency, with variant KD correlating with viral transduction. Capsid variants had a spectrum of phenotypes from no effect to an increase in transduction. This study represents the second report of the ubiquitously acting AAVR complexed with AAV capsids. However, while there is no major concerns with the context of this manuscript, it does not rise to the level for publication in Nature Communications following the publication of the AAV2-AAVR complex structure in Nature Microbiology (Zhang et al, Nature Microbiology 4, pages675–682 (2019)). This current manuscript reads like a continuation of this previous publication.

Response: Thank you for your comment to improve our manuscript. This work is not only a complementary to our previously work in Nature Microbiology, but also provide an example of a single receptor engaging multiple viral serotypes with divergent rules. These information will not only be

interested to AAV research field, but also advance the understanding of virus-receptor recognition in the virology field.

Minor Comments

Main text

Please run a spell check to correct typos (e.g. AA1)

Response: Sorry for the typo errors. We carefully checked the manuscript and correct the mistakes.

Lines 53-59 - Include reference to Dependoparvovirus group definitions in Cotmore et al - https://talk.ictvonline.org/ictv-reports/ictv_online_report/ssdna-viruses/w/parvoviridae/1043/genus-dependoparvovirus

Response: We include this citation.

Results

Line 120 - PKD2 also spans the 2/5 fold wall of the capsid

Response: We correct this point as suggestion.

Line 135, 198 – these are non-covalent interactions, or weak hydrophobic interaction

Response: We correct these two points as suggestion.

Line 141-143 – The AAV5 structure citation does not define secondary structure in this manner, cite the supplemental definition.

Response: We correct this point as suggestion.

Line 162-163 - Why is AAV2 used as control? Could be typo? Need AAV5. If this is AAV5, comparison with AAVR-AAV2 is not conclusive since AAV2 binds PKD2.

Response: Sorry for this typo error. It is AAV5. We remove the comparison with AAV2 in this revision.

Line 233 - Intermolecular bonds needs to be change to non-covalent interactions

Response: We change this point as suggestion.

Line 247 - Correct 3-G reference to residue K464 from K446.

Response: We correct K446 to K464 in Figure 3g.

Line 261 - AAV1 N383A not G

Response: Sorry for the mistake. We correct it to N383A.

Line 265 - Mutant and AAV1 needs to be swapped in sentence

Response: We correct this sentence.

Line 269 - not sure they have evidence for conformational alteration

Response: We remove this speculation and clarify that “The mechanism for this discrepancy needs further investigation”.

Comparison of the receptor interface on the AAV capsids- should keep nomenclature consistent, text is repeated from previous AAV2 paper mentioned above. This section is not really clear, and is most apparent in line 289-311

Response: As suggested by reviewers, we provide the comparison with SIA binding site with re-writing.

Line 282-283 - appears in previous AAV2-AAVR paper with a one word changed. Please rephrase

Response: We rephrase this sentence.

Line 287 - Missing R for VR

Response: We correct these errors.

Line 298 - Typo with residue numbers should be 585 and 588

Response: This mistake is corrected.

Line 301-302- Why use HSPG instead of sialic acid since both AAV1 and AAV5 bind sialic acid? Compare the glycan binding pockets for sialic acid with AAV1 and AAV5 - the information for these are published (fig 4). Huang et al, J Virol. 2016 May 12;90(11):5219-5230; Afione et al, J Virol. 2015 Feb;89(3):1660-72). Does the AAVR pocket overlap in anyway?

Response: As suggested by reviewers, we provide the information for the comparison with SIA binding sites in AAV1 and AAV5 in this revision. The SIA binding pocket on the AAV1 capsid has overlap with PKD2 binding region. The major SIA binding pocket (site A) on the AAV5 capsid, which play essential role for AAV5 transduction, is distant from PKD1 binding region.

Line 326 - P414 instead of P432 for supplementary figure 9

Response: This error is corrected.

line 347-348 typo- AAV5 instead of AAV2?

Response: AAV2 is the correct word here. We compare the different impacts of glycosylation of N395 on AAV2 and AAV5 transduction here.

Methods

Cell culture- no antibiotics mentioned

Response: line 393 “supplemented with 10% FBS” is rephrased to “supplemented with 10% FBS and 1% penicillin-streptomycin”.

Line 397 - need a product number for PEIMAX

Response: PEI MAX (Polysciences, 24765).

Lines 465-472 - The references, 27, 28 and 29 should be updated to the programs actually used

Response: We update the correct references.

Lines 475 - Why use AAV2 crystal structure instead of AAV1 – see RCSB pdb No. 3NG9.

Response: We change the initial model to the structure of AAV1 (PDB code: 5EGC) and cite the correct citation here.

Line 548 - Why overexpression (just expression)

Response: Line 548 “AAVR mutant overexpression and mCherry reporter assays”, is rephrased to “Ectopic expression of AAVR mutant and mCherry reporter assays”

Line 554 - 1000 ng instead of 100ng

Response: line 554-555 “A total of 100 ng or 1 µg plasmid was used for each well of the 96-well plate or 12-well plate” is rephrased to “A total of 100 ng plasmid was used for each well of the 96-well plate, or a total of 1000ng plasmid is used for each well of the 12-well plate.”

Supplemental

Supplemental figure 2 does not include residue 355

Response: We provide ^{AAVR}Y355 in Supplementary Figure 2a.

Supplemental fig 4; the body of the texts keeps mentioning PKD2 but it should be PKD1 since referring to AAV5 binding.

Response: Sorry for the mistakes. We checked all the text about the typos.

Response to Reviewer #2:

The manuscript by Lou and colleagues is focused on mapping divergent interactions of AAVR with different AAV serotypes, specifically AAV1/2/5 through cryoEM. The study clearly demonstrates that AAV5 interacts with AAVR through PKD1 and AAV1 does so by interacting with PKD2. The striking differences underlying the mechanics of how AAVR is recognized by two distinct AAVs using different surface domains/footprints for docking are a particularly novel finding in the current study. The study is carefully executed, technically sound, comprehensive in utilizing biophysical analysis as well as biological experiments (infectivity/transduction) and the manuscript is well written. Overall, the findings are novel, provide new insight into AAV biology - particularly, the study breaks new ground on how AAVR interactions might dictate AAV cell entry and transduction.

Response: We appreciate these comments.

Some concerns are outlined below.

First, are the mutations affecting AAV infection in a serotype specific manner or a global fashion? i.e., are AAVR PKD1/2 mutants able to rescue the non-cognate serotype? Specifically, do PKD1 mutants rescue AAV1 infection or PKD2 mutants rescue AAV5 infection? Second, what is the impact of AAV capsid mutations on binding to wt AAVR or their respective PKD domains? Only transduction data is shown (the term "Functional titer" in Figures 2H/3H needs to be better defined).

Response: We performed these experiments as request. As expected, the mutations on AAVR PKD1 do not affect the binding with AAV1 and the mutations on AAVR PKD2 do not affect the binding with AAV5 in Biacore experiments. And AAV1 presents the comparable infection in the AAVR-silenced cells with ectopically expressed PKD1 mutants and in the wt cells. AAV5 is the same. We provide these data in Supplementary Figure 9.

We also performed virus overlay assays to verify the impact of AAV capsid mutations on binding to wt AAVR. These data are shown in Supplementary Figure 8. The changes of the binding of AAVR and AAV capsid mutations are similar with the changes of virus transduction.

In our case, functional titer refers to the number of viral particles that can infect the cell in every μl . We use functional titer to evaluate the infectivity of mutant capsids. We define this in the figure legends.

Third, the results should be discussed against the backdrop of the glycan binding footprint on AAV1 and AAV5, both of which have been shown to bind N-linked sialic acid. Further, it is unclear how interaction of different AAVs with the same AAVR (albeit through different sites) can account for the divergent tissue tropisms observed. This should be highlighted as well.

Response: As suggested by reviewers, we provide the information for the comparison with SIA binding sites in AAV1 and AAV5 in this revision. The SIA binding pocket on the AAV1 capsid has overlap with PKD2 binding region. The SIA binding pocket (site A) on the AAV5 capsid, which play essential role for AAV5 transduction, is distant from PKD1 binding region. We also provide more descriptions in Discussion section.

Response to Reviewer #3:

This study is investigating the structural basis of the interaction between AAVR and different AAV strains. The authors determined four high resolution structures and showed that AAV1 binds to the second domain of AAVR (PKD2, similar to AAV2) whereas AAV5 binds to PKD1. Sequence alignments and structural comparisons were used to interpret the results.

The topic is of general interests and the findings are very inspiring. The manuscript is clearly written, and the figures are professionally prepared. Two important tasks behind the story are: 1. To show that indeed PKD1 is observed in the map of AAV5-AAVR; 2. To experimentally determine the roles of different amino acids that cause these distinct binding patterns. The authors have done good jobs addressing these issues.

Response: We appreciate these comments.

Here are a few specific comments:

1. Since the cryo-EM structures are the foundations of this manuscript, would it be possible to release the maps, either through the EMDB deposition system or private links?

Response: Yes, we have deposited the coordinate files and EM density to PDBank and EMDB with the accession numbers indicated in Table S1. According to the journal policy, we will release them before the formal acceptance of our work.

2. A lot of data are presented to show that AAV5 indeed binds to PDK1, but not much is discussed to depth. For example, what might be the biological advantage behind this phenomenon? Is there a particular reason why different AAVs evolved to recognize different regions on the same protein?

Response: We provide more descriptions in Discussion section.

3. In both cases of AAV1 and AAV5 complexes, a protein containing all five domains was used. Is there any trace of density coming from other domains (PKD1 and 3 for AAV1, PKD 2 for AAV5)? If not, why is the molecule so flexible?

Response: When we modulate the threshold to very low counter level, we could observe some untraceable density in the cryo-EM density. Because of the low occupancy, we cannot conclude whether these densities belong to AAVR or just are resulted by noise. Nevertheless, this indicates the other parts of AAVR do not display a fix conformation. Because AAVR is a typical multiple domain Ig-like protein, the flexible linkers between each PKD lead very flexible architecture of the extracellular part of AAVR. If one PKD does not contact with AAV capsid, it's orientation will be very variable.

4. The mutagenesis performed for AAV1-PKD2 and AAV5-PKD1 are mutually exclusive. Since a major point is being made comparing these two complexes, is it possible to see how mutations on PKD1 may affect the binding between AAVR and AAV1, and how mutations on PKD2 affect the binding between AAVR and AAV5? Results on a couple of mutants would be good enough.

Response: We performed these experiments as request. As expected, the mutations on AAVR PKD1 do not affect the binding with AAV1 and the mutations on AAVR PKD2 do not affect the binding with AAV5. We provide these data in Supplementary Figure 9.

5. Mutations on AAVRT350 and AAVRR353 abolished binding to AAV5, but the infectivity was not much influenced, why?

Response: We appreciate this comment. Honestly say, we do not have solid evidence to draw a conclusion for this discrepancy, but the data is indeed experimentally here. We speculate the discrepancy is caused by that the AAVR protein used for Biacore experiments is a recombinant protein and its

conformation might have slight difference compared with it is on the cell surface.

6. Are the clashing scores calculated for the entire capsids? Was icosahedral symmetry applied in the model refinements? What are the correlation coefficients between the maps and the models (such value is usually reported by phenix real space refinement)?

Response: Yes, we applied the icosahedral symmetry for the 3D classification and 3D refinement. Thus, there is no clashing between adjacent capsomers. We also provide the correlation coefficients between the maps and the models in Supplementary Table 1.

7. It would be helpful to label residues that are conserved between AAV1, 2 and 5 in supplementary tables 3 and 5.

Response: We tried to label the conserved residues between AAV1, 2 and 5 in Supplementary Table 3 and 5, but it made the tables too complicated, since the residues conserved in AAV1/2, AAV2/5, AAV1/5 are not the same. Alternatively, Supplementary Figure 10 is clear to observe the residue alignment among all AAV serotypes. We hope reviewer 3 could understand our concern and allow us to keep the current version of Supplementary Table 3 and 5.

8. Two potential typos: In line 230 and 749, should it be “PKD2” instead of “PKD1”? The caption of sup figure 6 says “supplementary figure 4”.

Response: Sorry for this mistake. We correct “PKD1” to “PKD2”. And the caption of Supplementary figure 6 is corrected.

Reviewers' Comments:

Reviewer #1:

Remarks to the Author:

The authors have responded adequately to previous review comments.

Reviewer #2:

Remarks to the Author:

The authors have substantiated their major claims that divergent AAV capsids engage AAVR in different ways through further experimentation. All concerns/questions raised by this reviewer have been addressed. The study is technically sound, the structural data are of high quality and the new insight into virus-receptor engagement should be well received by the broad readership of this journal. Further, the study is likely to pave the path for studying and manipulating AAV biology and consequently influence future applications in gene therapy.

Reviewer #3:

Remarks to the Author:

I find the revised manuscript satisfactory and recommend it for publication.